# Tibetan Plateau grasslands might increase sequestration of microbial necromass carbon under future warming
Qinwei Zhang[1,2,6], Xianke Chen[1,2,6], Xiaorong Zhou[1,2], Xin Nie[1,2], Guohua Liu[1,2], Guoqiang Zhuang[1,2,3], Guodong Zheng [4], Danielle Fortin[5] & Anzhou Ma [1,2]✉

Microbial necromass carbon (MNC) can reflect soil carbon (C) sequestration capacity. However, changes in the reserves of MNC in response to warming in alpine grasslands across the Tibetan Plateau are currently unclear. Based on large-scale sampling and published observations, we divided eco-clusters based on dominant phylotypes, calculated their relative abundance, and found that their averaged importance to MNC was higher than most other environmental variables. With a deep learning model based on stacked autoencoder, we proved that using eco-cluster relative abundance as the input variable of the model can accurately predict the overall distribution of MNC under current and warming conditions. It implied that warming could lead to an overall increase in the MNC in grassland topsoil across the Tibetan Plateau, with an average increase of 7.49 mg/g, a 68.3% increase. Collectively, this study concludes that alpine grassland has the tendency to increase soil C sequestration capacity on the Tibetan Plateau under future warming.

The Tibetan Plateau is more sensitive to climate change than other regions due to its high altitude, many glaciers, and abundant soil organic carbon (SOC) storage, among which SOC can regulate climate by restoring carbon (C) sinks and preventing further $CO_2$ emissions[1]. Consequently, SOC sequestration in the Tibetan Plateau's terrestrial ecosystems is critical in mitigating climate change. As an important contributor to persistent soil C pool, microbial necromass carbon (MNC) accounting for up to 30%–87% of SOC[2]. One way that MNC is thought to accumulate is through the production and stabilization of microbial residues while achieving long-term sequestration in the soil due to its mineral stabilization[3]. As such, MNC can play an integral role in maintaining and potentially enhancing the sequestration of SOC in ecosystems like the Tibetan Plateau. Therefore, estimating the overall changes in the MNC under warming climate conditions can help us explore the variations in the soil C sequestration capacity and C storage in the Tibetan Plateau under climate change.

Previous studies have proven that the mean annual temperature (MAT) indirectly affects MNC by affecting the aridity index (AI) and net primary productivity (NPP)[4], and the plant C input and mineral protection are the most important driving forces of the MNC in the surface soil of alpine grasslands[5]. Emerging research revealed that some complex controls on MNC accrual and its role in long-term SOC storage. For example, warming can accelerate microbial metabolism, potentially increasing microbial turnover rates and necromass carbon accrual[6]. However, warming may also lead to increased decomposition rates of SOC, thereby posing potential challenges to the storage of SOC in warmer future scenarios[7]. With regard to the impact of warming on MNC, most previous research has been limited to small-scale experiments at field experimental stations, especially the influence of warming on the distribution of MNC on the Tibetan Plateau. Liang et al.[8] observed a significant decrease in the quantity of microbial residue during a 9-year warming experiment in a California grassland. Ding et al.[9,10] found that warming increased the contribution of microbial residue to the SOC in the 0–50 cm soil layer in an alpine meadow. Jia et al.[11] found that the quantity of $^{13}C$-microbial residues used by topsoil microorganisms after warming did not significantly change. It can be seen that the complex effects of multiple environmental factors on the C sequestration processes mediated by the necromass which resulted in uncertain distribution results of MNC, especially the evaluation of regional scale MNC is seriously insufficient. Therefore, a comprehensive understanding of the overall MNC distribution on the Tibetan Plateau under warming is of great significance for comprehending soil C dynamics under regional warming.

Here we conducted sampling across a span of ~20° longitude (Fig. S1) and estimated the MNC by measuring amino sugars[2,12]. Subsequently, we

[1]Research Center for Eco-Environmental Sciences, Chinese Academy of Sciences, Beijing 100085, China. [2]University of Chinese Academy of Sciences, Beijing 100049, China. [3]Binzhou Institute of Technology, Binzhou 256600, China. [4]School of Environmental Studies, China University of Geosciences, Wuhan 430078, China. [5]Department of Geology, University of Ottawa, Ottawa K1N6N5, Canada. [6]These authors contributed equally: Qinwei Zhang, Xianke Chen. ✉e-mail: azma@rcees.ac.cn

created a comprehensive MNC dataset based on amino sugars. To determine the climatic, plant-related, and soil physicochemical indexes of the samples, we used a combination of laboratory measurements and an environmental factor database. 16S rRNA gene data[13,14] was used to predict the relative abundances of the dominant eco-clusters[15]. This study pioneered the use of a quantitative index combining environmental factors and microbial communities to explore its predictive effect on MNC. We further ascertained the soil's capacity for C transformation by quantifying the expression of genes related to C degradation and fixation, employing quantitative polymerase chain reaction (qPCR) as our measurement method. Afterward, we incorporated multiple factors into a deep learning model to predict the overall distribution pattern of the MNC in the Tibetan Plateau under the current conditions and warming. We aimed to focus on quantifying how MNC stocks change in response to warming across the region.

## Results

### Associations of spatial MNC distribution and soil microbial properties

Across the Tibetan Plateau, the MNC in the alpine grassland topsoil exhibited distinct spatial patterns from east to west (Fig. 1). The MNC in the topsoil was 0.2–53 mg/g, with an average of 8.8 mg/g (Fig. S2). Among them, the MNC of the alpine meadows (average of 14.1 mg/g) was generally significantly higher than that of the alpine steppes (average of 3.9 mg/g) (Fig. S2). The microbial biomass based on the 16S rRNA gene concentration in the alpine meadows was less than $3 \times 10^7$ copies/ng, which was generally lower than that in the alpine steppes (Fig. 1). Interestingly, a marginal negative correlation was observed between the MNC and microbial biomass ($R^2 = 0.41$, $p < 0.01$) (Fig. 1). C fixation and degradation genes had a secondary correlation with MNC in the topsoil, and there is no clear rule between C circulating gene and MNC (Fig. 1). Besides, using PC1 to represent the beta diversity of the microbial community, it was found that PC1 was significantly positively correlated with MNC (Fig. S3), and the fitting effect was better ($R^2 = 0.42$, $p = 0.0004$), indicating that the dominant microbial community has an important contribution to MNC.

Based on the habitat preferences of the dominant phylotypes, seven bacteria eco-clusters and three fungi eco-clusters were defined. The seven bacteria eco-clusters were as follows: high elevation (Hele), high elevation

and low normalized difference index (Hele&LNDVI), high silt (Hsilt), high mean annual precipitation (HMAP), high total nitrogen (HTN), HMAP and low MAT (HMAP&LMAT), and high SOC and low TN (HSOC<N). The three fungi eco-clusters were as follows: high NDVI and low NPP (HNDVI&LNPP), high NPP and low NDVI (HNPP&LNDVI), and low thickness (Lthickness) (Fig. S4). The ten eco-clusters had diverse taxonomic compositions at the bacteria phylum and fungi class level (Fig. 2). Fungi phylotypes were more likely than bacteria phylotypes to co-occur with other phylotypes belonging to the same eco-clusters (Fig. 2). In addition, the coexistence of phylotypes between different bacteria eco-clusters was more likely to occur (Fig. 2). The Cubist model was used to predict the various relative abundance distribution of each eco-cluster based on the different environmental preferences of the phylotypes (Fig. S5).

### Model structure and validation

Random forest (RF) analysis revealed that the NDVI was the most important variable, followed by the HMAP&LMAT relative abundance. Hele relative abundance, sand content, and silt content contributed roughly equally to explaining the variations in the MNC. The variables ranked after TN were considered to be variables with no contribution to the MNC. Overall, the relative abundances of the bacteria eco-clusters had a more significant effect on the MNC than that of the fungi eco-clusters (Fig. 3a). Based on the RF results, the top 19 variables in terms of importance were selected as the input variables of the model. The models with input variables containing the relative abundances of the dominant eco-clusters were labeled CE, and those without were labeled WE. The CE model included all 19 variables, while the WE model included only 14 variables and did not contain the relative abundance of HMAP&LMAT, Hele, HSOC<N, Lthickness, and Hele&LNDVI. The K-fold cross-validation results show that the CE model has a better simulation performance, with a higher $R^2$ (0.82) and lower root mean square error (RMSE = 4.16) than the WE model ($R^2 = 0.75$, RMSE = 4.91) (Fig. 3b, c; Table S1). The hold-out validation of the CE model shows that the MNC observations are significantly correlated with the fitted values, with a slope of 0.97 and a Pearson's correlation coefficient of 0.917 ($p < 0.001$; Fig. 3d). The hold-out validation of the CE model is better than that of the WE model (Fig. 3d, e). The above conclusion further proves that the relative abundance of the dominant eco-clusters plays a vital role in the prediction of the MNC.

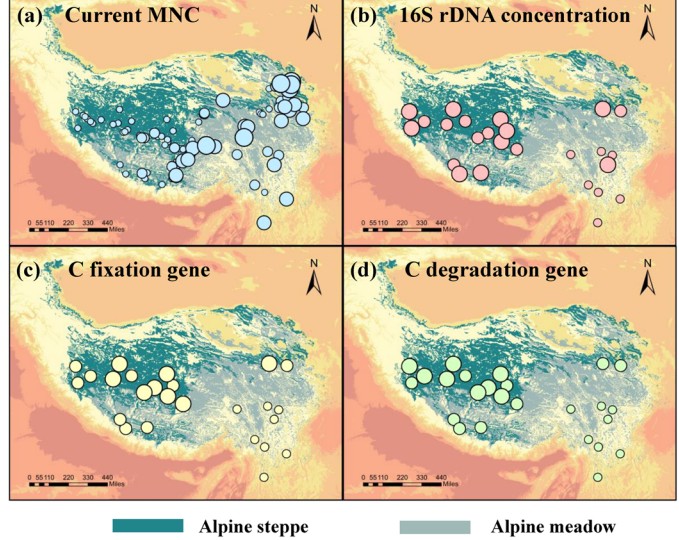

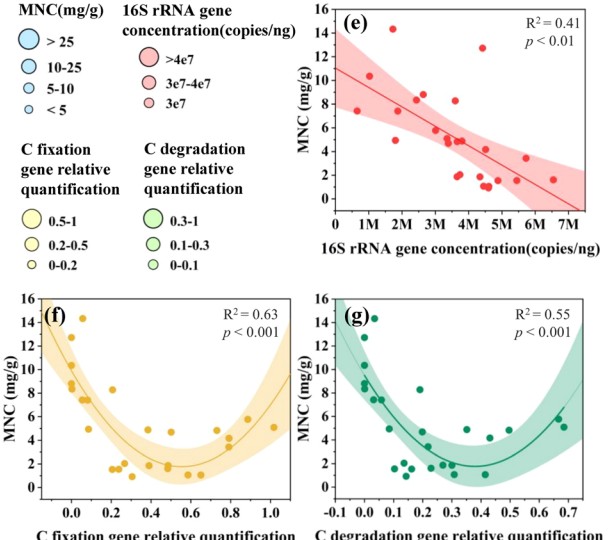

**Fig. 1 | Relationship between MNC and microbial related genes in the Tibetan Plateau. a** Spatial distributions of MNC ($n = 96$), (**b**) 16S rRNA gene concentration ($n = 25$), (**c, d**) C fixation and degradation gene, and (**e–g**) their association with the current MNC in the topsoil across the Tibetan Plateau alpine grassland ($n = 25$). The

M in the x-axis in (**e**) represents a million. The relative quantification of (**f**) and (**g**) represents the ratio of gene relative quantification to 16S rRNA gene relative quantification.

## Spatial distribution of MNC currently and under warming

Based on the MNC simulation results of the CE model, we mapped the geographic distribution of the MNC in the Tibetan Plateau (Fig. 4a). The MNC is high in the northeast Tibetan Plateau (highest value of

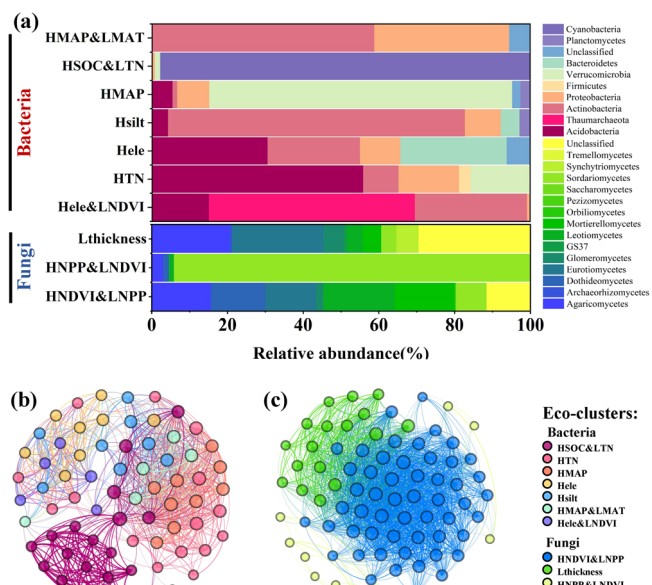

**Fig. 2 | Abundances and compositions of defined eco-clusters and their network of interactions. a** Percentage of OTUs in each bacteria eco-cluster at the phylum level and fungi eco-cluster at the class level (n = 172). HMAP&LMAT: high MAP and low MAT, HSOC<N: high SOC and low TN, HMAP: high MAP, Hsilt: high silt, Hele: high elevation, HTN: high TN, Hele&LNDVI: high elevation and low NDVI, HNDVI&LNPP: high NDVI and low NPP, Lthickness: low thickness, HNPP&LNDVI: high NPP and low NDVI. **b** Bacteria network diagram with bacteria phylotypes as nodes and their Spearman correlation coefficient as edges. **c** Fungi network diagram with fungi phylotypes as nodes and their Spearman correlation coefficient as edges. The density of the node aggregation represents the proximity of the phylotypes. Each node represents a phylotype, and each color represents an eco-cluster.

43.73 mg/g) and is low in the west (lowest value of 0.6 mg/g). The MNC in the Tibetan Plateau alpine grasslands exhibits a positive skewness distribution, with an average of 10.96 mg/g (Fig. 4d). The distribution is similar to that of the 96 observation sites. The contributions of the MAT and MAP to the relative abundances of the eco-clusters evaluated using the Cubist model shows that the dominant eco-clusters added to the CE model are sensitive to changes in temperature and precipitation, except for the Lthickness without the variable explanation of the MAT and MAP (Table S2). The Hele had the lowest sensitivity, so the influence of warming on it can be neglected. The independent-sample Mann–Whitney U test of the site data revealed significant variations in the relative abundances of the temperature-sensitive eco-clusters over the entire Tibetan Plateau under RCP8.5 in the 2050 s compared to the present (p < 0.05) (Table S3). According to the results of the RF analysis, MAT has little direct effect on the MNC. However, the distribution of the MNC throughout the entire Tibetan Plateau exhibits apparent changes under RCP8.5 in the 2050 s vs. current. The hotspot of high MNC remains in the eastern area, but the scale of the hotspot expands under warming (Fig. 4a, b). Overall, MNC increases in the entire Tibetan Plateau, but the rate of increase is more considerable in the alpine meadow than in the alpine steppe (Fig. 4c). The relative abundance of MNC and HMAP&LMAT in the meadow are positively correlated (Fig. S6). The average increases in the MNC in the alpine grassland and the alpine meadow are 6.34 mg/g and 8.68 mg/g, respectively (Fig. 4d). The mean MNC value increases from 10.96 mg/g to 18.45 mg/g under simulated warming conditions (Fig. 4d).

## Discussion

The microbial uptake of plant-derived C affects the consumption and accumulation of MNC. The quantitative results illustrated that alpine grassland soil with a high MNC value tended to have a low bacterial biomass (Fig. 1). However, the high degree of expression of C degradation genes had no significant effect on the accumulation of MNC (Fig. 1). In terms of MNC generation, we believe that it is not because a larger microbial biomass is more conducive to the accumulation of MNC; instead, a low microbial biomass microbiome with a short generation time and survival time may lead to the accumulation of MNC by means of consistently producing microbial

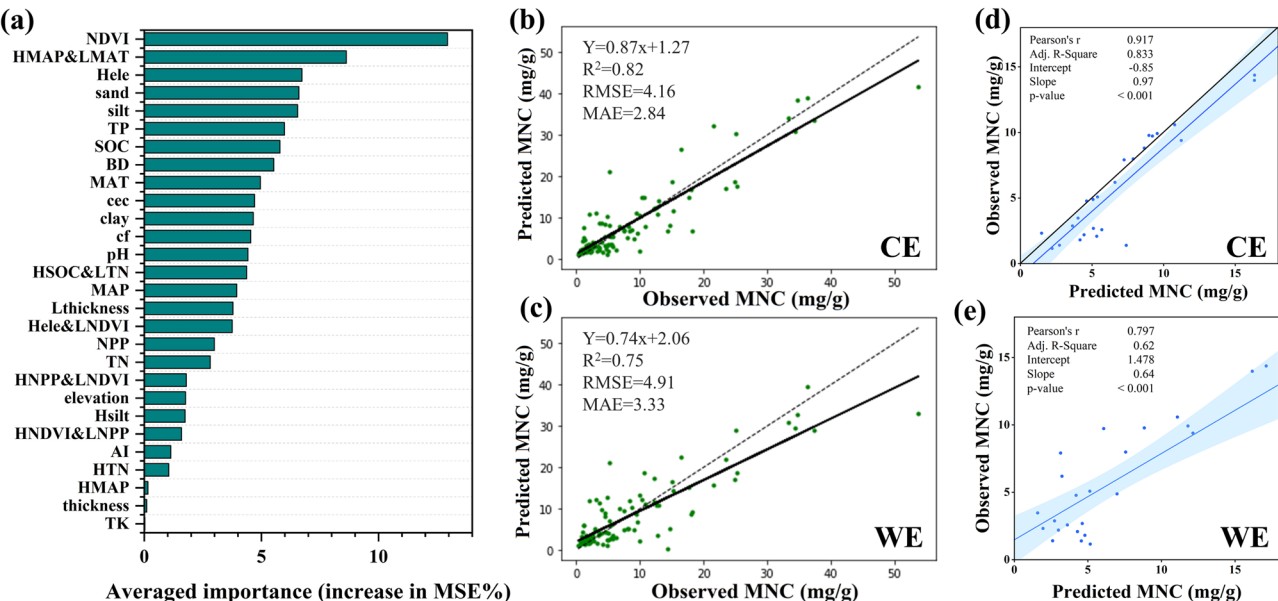

**Fig. 3 | RF analysis results and model validation. a** Importance ranking of RF analysis between MNC and environmental variables (n = 96). **b** CE model and (**c**) WE model K-fold cross validation result (n = 96). **d** CE model and (**e**) WE model hold-out validation result (n = 25). The degree of the linear fit of the points in (**d**, **e**) shows the relationship between observed and predicted MNC on a linear scale.

RMSE root mean square error, MSE mean square error. NDVI normalized difference vegetation index, TP total phosphorus, BD bulk density, MAT mean annual temperature, cec cation exchange capacity, cf gravel content greater than 2 mm, MAP mean annual precipitation, NPP net primary productivity, TN total nitrogen, AI aridity index, TK total potassium.

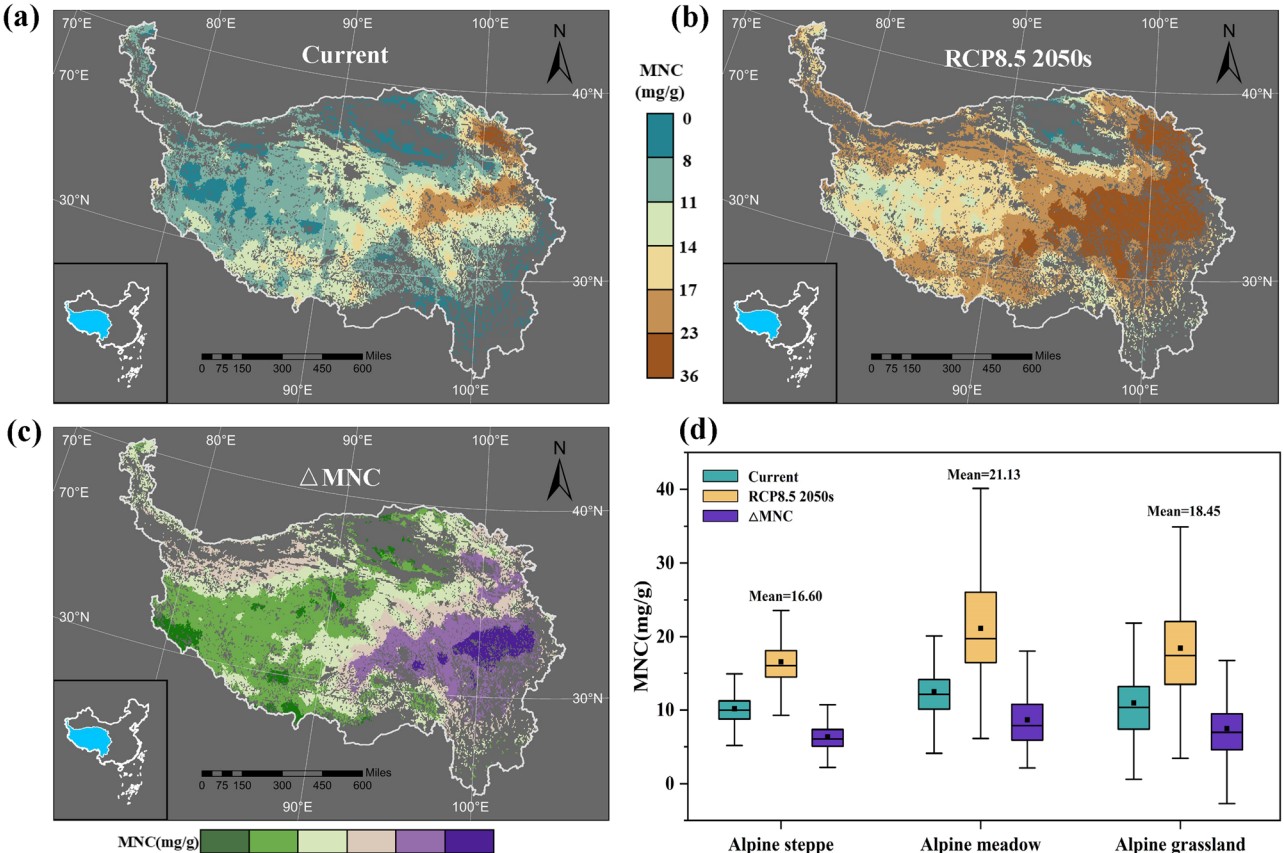

**Fig. 4 | Projected distribution of topsoil MNC in grasslands across the Tibetan Plateau. a** Current, (**b**) under RCP8.5 in the 2050 s, and (**c**) their D-values distribution. **d** Comparison of topsoil MNC in the alpine steppe ($n = 4717$), alpine meadow ($n = 6033$), and alpine grassland ($n = 10,157$) (current *vs.* RCP8.5 in 2050 s) and their D-values. The alpine grassland map was derived from China's Vegetation Atlas (Editorial Committee for Vegetation Map of China, 2001). The grassland consists of meadows and steppes. The horizontal line and square black dots in each box represent the median and mean, respectively. The different colors in the box line chart denote the different periods (green: current, yellow: under RCP8.5 in 2050 s), and purple represents their D-values.

residues[16]. The metabolism of a microbial community (rate of growth, consumption of nutrients, etc.) is dependent on its composition, which includes species diversity, relative abundance, and the functional traits of the species present[17]. From the perspective of MNC consumption, the decomposition rate of MNC varies between microbial taxa[18]. In summary, we found that a single microbial variable, such as microbial biomass and C cycling gene, could not explain how the MNC accumulation proceeds. Therefore, we further explored the joint effect of microbial and environmental variables on MNC through an integrated concept, eco-clusters[15].

Four bacteria and one fungi eco-clusters were found to substantially affect the MNC using RF analysis (Table S2). The HMAP&LMAT eco-cluster mainly consists of Actinobacteria and Proteobacteria, both of which are the main components of the microbial necromass in the soil[18]. Acidobateria and Bacteroidetes are the typical taxa that follow oligotrophic and copiotrophic lifestyles[19], occupying more than half of the Hele. Cyanobacteria account for more than 99% of the HSOC<N, and Hele&LNDVI even contains more archaea. Collectively, the classification of taxa based on habitat preference embodies the effect of the environmental factors and contains the survival strategies of the microbial composition. RF analysis revealed that some climate factors, such as the MAT, alone do not significantly impact the MNC. However, these environmental factors indirectly affect the MNC by adjusting microbial community compositions[20]. We also observed that the effect of the bacterial community on the MNC was greater than that of the fungi. Previous studies have proven that the replacement of cell wall components and the metabolic rate of bacteria occur more rapidly than that of fungi[21,22], and bacterial decomposition is more responsive to changes in nutrient availability and may thus play a larger role

in the C cycle[23]. We speculate that this is because the active renewal of bacterial cell walls can amplify the bacterial turnover based on the habitat preference, resulting in the bacteria having a greater effect on the MNC than the fungi.

RF analysis shows that the variations in the MNC in the different latitude ranges and regions of the Tibetan Plateau result from the direct and indirect effects of climate change. More minor changes could indicate a better buffering capability at the soil level. Significant changes in the relative abundances of the dominant eco-clusters, will occur by the 2050 s under RCP8.5 when compared with those at the present time (Table S3). A likely explanation for this is that climate warming increases the soil buffering capacity by indirectly influencing the microbial community changes (Table S3), specifically the adaptability of the microbes to warming[24]. Several studies have reported results similar to our overall prediction of the MNC in the alpine grassland topsoil on the Tibetan Plateau under climate warming conditions. After warming, the net increase in the microbial residues in the surface soil is 23.9%[9], and warming significantly increases the microbial residues in marsh meadows[10]. Previous studies have proven that warming of surface soil increases the plant-derived C input and the available microbial resources[25]. Microorganisms tend to consume more organic carbon for their anabolism, which is more conducive to the accumulation of microbial residues and markedly increases the proportion of residual microbial carbon in the organic carbon pool in the grassland on the Tibetan Plateau[9].

An interesting finding was that the MNC will increase more in the meadow than in the steppe under warming conditions. As has been previously reported, the MNC is significantly higher in the alpine meadows than in the alpine steppes[5], which was also demonstrated in our study

(Fig. 4d). Ding et al.[9] reported that the increase in the MNC in the alpine meadow was related to the increase in the microbial turnover rate under a warming background. In addition, we found that there were notable differences in the dominant phylotypes abundance between the meadow and steppe. We speculate that the geographic differences in the distribution patterns of the dominant phylotypes lead to regional differences in the microbial turnover, which further results in differences in the MNC growth rate in the two types of alpine grassland under warming conditions (Fig. S5). Based on the above conclusions, we propose that the dominant phylotypes in the alpine meadow are more sensitive to warming, and the eco-clusters adjust their adaptability to warming by changing the relative abundances of the components of the community, suggesting that the abundances of the dominant phylotypes can be used as an indicator to assess and predict alpine grassland microbial C storage under climate change.

The AI is significantly different in alpine meadows and steppes, with a boundary threshold of about 0.27[26]. Aridity has an essential effect on the accumulation coefficient of MNC in grasslands, which reflects the direct and indirect effects of soil moisture and precipitation on MNC[4]. Soil moisture is considered to be a critical factor driving the accumulation of MNC under warming conditions[27]. The higher humidity of the soil in the alpine meadow (AM) compared to that in the alpine steppe (AS) results in higher N and phosphorus (P) utilization rate of the plants and thus a higher soil nutrient homeostasis[28]. Fan et al.[29] proved that the stoichiometric stability of the soil microbial biomass is more stable in AM than in AS through large-scale sampling measurements of MBC and microbial biomass nitrogen (MBN). Fungal residues accounted for a higher proportion of the SOC than bacterial residues[2], accumulating more easily in acidic soils[4,30]. The pH of AM soil with a higher AI is lower than that of AS soil throughout the entire Tibetan Plateau[26]. Therefore, the soil environment of the AM is more conducive to the accumulation of MNC. In addition, the vegetation coverage of AM (NDVI > 0.6) is generally higher than that of AS (NDVI < 0.4), and there is a significant positive correlation between NDVI and MNC (Fig. S7). In conclusion, regarding the hydrothermal conditions, soil homeostasis, plant-derived C input, and microbial survival adaptability of the microbes in the AM are more conducive to biological C sequestration than those of the microbes in the arid environment of the AS and thus achieve effective, long-term C sequestration in the soil. Warming will accelerate soil C loss[31–33], so the regional soil MNC accumulation in the Tibetan Plateau is considered to be a critical natural solution for mitigating warming. We speculate that the tremendous increase in the MNC in the AM due to warming is a protective strategy to increase soil C sequestration and reduce C loss in the Tibetan Plateau in order to mitigate the negative impacts of climate change.

Due to the lack of large-scale systematic observations of the MNC in the Tibetan Plateau, we collected published MNC data from 71 sites[4,5,9,34–37] and combined them with our site measurements to create the training set of the model. In addition, it was inevitable that we could not obtain some critical factors that have been proven to directly influence the MNC in previous studies, such as mineral protection related indicators[38], actual soil moisture data[39], and aboveground and belowground C input from plants[5,40]. In order to make up for the loss of these direct influencing factors, we adopted the methods of relevant factor replacement, multi-factor interaction, and construction of new comprehensive indicators, combined with the powerful multi-variable learning ability of deep learning, to achieve the current simulation effect (Fig. 3d). The mineral protection degree of soil microbial residues is closely related to the soil particle size, and soil microaggregates also affect the response of microbial physiological activities to climate change[41]. Besides, MAP and AI were selected to reflect the precipitation and drought conditions. We have proven that this is due to the regional dividing line of the MNC distribution under the influence of water and heat conditions. The deep learning model can effectively capture the strength of the nonlinear correlation, thereby weakening the influence of the boundary threshold[42,43]. Besides, the process of ecological cluster division includes the selection of dominant phylotypes based on hydrothermal conditions and regional differences in their abundances under the influence

of mineral conservation[15], further indicating the distribution of the MNC from the perspective of microbial adaptability.

In summary, the constructed CE model emphasizes the original driving role of the microorganisms in MNC accumulation process, which is different from previous models that ignored the role of microorganisms. The simulation results of the current model are consistent with the actual situation and have specific guiding significance for changes in soil C storage in the Tibetan Plateau and also provide ideas for incorporating microbial effects into the soil C cycle model. Even though the existing datasets can achieve the desired effect, more MNC measurements are needed to further improve the simulation performance of the model.

## Methods
### Sampling sites description
We collected samples between 2019 and 2021 from west to east across about 20° of longitude in the Tibetan Plateau (Fig. S1). Six composite samples were collected at each site in the four corners and the middle area of a 10 m × 10 m quadrat. Soils (0–15 cm) were collected and were sieved through a 2-mm sieve to remove impurities such as stones and plant roots. Then, soils were bagged and stored in an ice box. The remaining 147 samples were collected along a transect spanning ~3500 km across the Tibetan Plateau during 2013–2014, and the specific sampling method has been described by Ding et al.[13]. Later, the observation of MNC from 71 samples were obtained from seven papers[4,5,9,34–37] by searching for the terms "Tibetan Plateau", "amino sugar" and "microbial residue carbon" on the Web of Science (http://apps.webofknowledge.com/). Meadows and steppes account for most of the alpine grasslands on the Tibetan Plateau, with alpine meadows distributed in the wetter and warmer southeastern region and grasslands in the drier northwestern region. The site coverage was relatively uniform, including 46 meadow and 50 steppe sample sites[4].

### Sequencing data collection and analysis
The eDNA was extracted using a MinkaGene Soil DNA Kit (Guangzhou mCHIP BioTech CO). The concentration and purity were measured using NanoDrop One (Thermo Fisher Scientific, MA, USA). The 16S rRNA gene amplification primers targeted in the V4 hypervariable region included 515 F (5'-GTGCCAGCMGCCGCGGTAA3') and 806 R (5'-GGAC-TACHVGGGTWTCTAAT-3')[44]. ITS3 (5'-GCATCGATGAAGAACGCA GC-3') and ITS4 (5'TCCTCCGCTTATTGATATGC-3') were used to amplify the fungi ITS2 gene[45]. PCR amplification, sequencing library construction, library quality evaluation, and sequencing are described previously[46]. All of the sequence analyses were performed using the Galaxy pipeline (http://mem.rcees.ac.cn)[47].

### Characterization of functional genes related to carbon
The total amount and purity of the DNA were measured using a Qubit 4.0 (Thermo Fisher Scientific, Waltham, USA) instrument to ensure that the DNA concentration was uniformly diluted in the 20 ng/μl. The representative bacterial biomass was obtained by measuring the 16S rRNA gene concentration[48]. Absolute quantitative information of 16S rRNA gene was obtained by fluorescence quantitative PCR (Roche, LightCycler480 II). The qualified DNA samples were added to a 384-well plate (the sample sourceplate), and the primer and quantitative PCR (qPCR) reagent were added to another 384-well plate (the assay sourceplate). A SmartChip Multisample Nanodispenser (Takara Biomedical Technology) was used to add the sample sourceplate and assay sourceplate reagents to the micropores of the SmartChip MyDesign Chip (Takara Biomedical Technology, Clontech) and a high-throughput qPCR chip, respectively. The qPCR reaction and fluorescence signal detection were performed using a SmartChip Real-Time PCR System (WaferGen Biosystems USA), and the amplification curve and dissolution curve were automatically generated. According to the Ct values of each gene in each sample given by SmartChip Real-Time PCR System and Canco software, the relative quantitative information of each gene in each sample was calculated according to the relative quantitative = 10(31−Ct)/(10/3) formula after the above quality control. Only the genes detected in the

three technical repetitions were judged to be positive, and the average value was calculated as the relative quantification of the gene in the corresponding samples.

## Measurement and collection of environmental factors contributing to MNC

To investigate the degrees of influence of the various environmental factors on the MNC, we collected databases of climate, plant, and soil physicochemical indicators that included data for the geographical coordinates of each site. The climatic variables (MAT and MAP) were obtained from the WorldClim database (http://www.worldclim.org) for 1982–2016, and the aridity index (AI) was obtained from the CGIAR-Consortium for Spatial Information (CSI) GeoPortal (https://cgiarcsi.community) for the period 1970–2000. We used the multi-year averages of the climate variables to explore their effects on the MNC. The plant variables (NPP and NDVI) were downloaded from the National Aeuronatics and Space Administration (NASA) Earth Observations website (NEO, https://neo.sci.gsfc.nasa.gov/), and we selected the average NPP and NDVI for the sampling year.

Soil physicochemical indicators, including SOC, TN, TP, and pH, were measured according to previous work[46]. Before analysis, all soil samples were air-dried, ground, and processed through a 0.15-mm mesh sieve. The other soil property indicators, including total potassium (TK), silt concentration (silt)[48], clay concentration (clay), sand concentration (sand), gravel concentration (cf), soil bulk density (BD), soil thickness (thickness), and cation exchange capacity (cec) were obtained from the National Earth System Science Data Center, National Science & Technology Infrastructure of China. (http://www.geodata.cn). All of the downloaded raster data with a spatial resolution of 1 km were extracted in ArcGIS 10.7 according to the specific geographic coordinate information.

## Analysis of amino sugars

The soil MNC content was determined by measuring the amino sugars in the soil. The extraction and determination of the soil amino sugars were conducted according to the method described by Indor and Mou[49,50]. Briefly, 0.5 g of air-dried soil sample was mixed with 10 ml of 6 M hydrochloric acid and was hydrolyzed at 105 °C for 6 h. The cooled hydrolysis solution was blown dry using nitrogen at 30 °C to evaporate the excess hydrochloric acid. Then, the remaining hydrolysis product was dissolved in 2 ml of ultrapure water after filtration and was stored at 4 °C. The hydrolysis products were derivatized online using ortho-phthaldialdehyde (OPA) and were separated using a Hypersil GOLD C18 column (Acclaim120 C18; 4.6 mm × 150 mm, 3 μm; Thermo Fisher Scientific, Waltham, USA) at 35 °C. Four amino sugars (Glucosamine (GluN), galactosamine (GalN), muramic acid (MurA), and mannosamine (ManN)) were determined and analyzed using a high-performance liquid chromatograph (Dionex Ultimate 3000, Thermo Fisher Scientific, USA) with emission and excitation wavelengths of 445 nm and 330 nm. The total microbial residual carbon was calculated by combining the conversion coefficients of the bacterial and fungal residual carbon with the amino sugar fraction[2].

## Identification of ecological clusters

The microorganisms were divided into ecological clusters based on habitat preferences according to the method of ref. 15. Furthermore, the relative abundance of each ecological cluster was calculated. First, the top 10% of the phylotypes in terms of abundance and presence in more than half of the sample size were selected from overall operational taxonomic unit (OTU) analysis[51], and named as dominant phylotypes. Secondly, the extracted dominant phylotypes were combined with all of the environmental variables using the random forest model analysis to screen out the phylotypes with habitat preferences with variable explanations of ≥30%. Thirdly, the ecological clusters were identified using semi-partial Spearman correlation and clustering analysis, and the relative abundance of each ecological cluster was calculated separately. The relative error magnitude should be less than 1, which was used to evaluate the Cubist model fits. Finally, the relative abundance of each ecological cluster was predicted using the Cubist model and the environmental variables. Their distribution in the Tibetan Plateau region was then mapped using the kriging function in ArcGIS 10.7.

## Model structure and validation

Random forest analysis was used to screen the input variables of the model. The top 70% of the environmental factors were selected as the input variables. The grassland MNC was estimated using stacked autoencoder networks[43], and the model structure was divided into four layers: input layer, autoencoders (AEs), regressor, and output layer. The input layer was the target variable MNC and the screened environmental variables. The AEs were used to generate the model, and its inputs were reconstructed by extracting the high-level features. The regressor used a neural network to make predictions using the features extracted by the AEs. The output layer output the predicted MNC. The model first underwent layer-by-layer unsupervised pre-training, that is, the environmental variables were input, the AEs were trained, the high-level features were extracted from the environmental variables, and the model weights were initialized. After several model structure adjustments and parameter optimization, the AEs adopted a three hidden layer structure, including two compression layers and one release layer. The compression layer compresses the neurons one latitude at a time, and the release layer releases them one latitude at a time. The artificial neural network (ANN) in the stacked autoencoder (SAE) contains a hidden layer with 32 neurons. The model was then fine-tuned by inputting the environmental and target variables and fine-tuning the model weights. K-fold cross-validation and hold-out validation were used for the model validation. Among them, K-fold cross-validation adopts tenfold cross-validation. The input dataset was equally and randomly divided into 10 subsets. Nine were used as training sets, and one was used as the validation set. The hold-out validation dataset used 25% of the sample observations we set aside in advance and did not use in the model training. The model predictions were linearly fitted to the observations; and the slope, Pearson's r value, $R^2$, root mean square error (RMSE), and mean absolute error (MAE) were selected as the model evaluation metrics. All of the above steps were implemented in Python (3.8.5).

## Prediction under future climate scenarios

The climate scenario was RCP8.5, in which the global mean temperature rises to 5 °C by 2100 relative to preindustrial times. This scenario was used to simulate the variation in the MNC in the 2050 s under climate warming conditions. Based on the previously established Cubist model, we replaced the climatic variables with the relevant parameters of RCP8.5, and the soil physicochemical properties were kept consistent with the current values. We derived the relative abundances of the dominant ecological clusters under the RCP8.5 future scenario. The Mann–Whitney U test was conducted on each eco-cluster separately to investigate the differences in the relative abundances of the dominant ecological clusters before and after warming. The future climate projections under RCP8.5 in the 2050 s were derived from climate model BCC-CSM1.1 (originated from Beijing Climate Center, China), which was downloaded from the climate change, agriculture and food security (CCAFS)-Climate data portal (http://www.ccafsclimate.org/dataspatialdownscaling/). The predicted relative abundances of the dominant ecological clusters and the related climatic parameters under RCP8.5 in the 2050 s were jointly input into the established CE model to output the predicted MNC under warming. Finally, the distribution pattern of the MNC was visualized using ArcMap 10.7.

## Statistics and reproducibility

The correlation between the two groups was analyzed using regression analysis. Data was considered statistically significant if $p < 0.05$. Random forest analysis was performed using the R package 'ppcor' and visualized by R package 'pheatmap' in R 3.5.1. R packages 'Cubist', 'gstat', 'raster', 'sp', 'maptools' and 'ggplot2' were used for model building and map visualization in R 3.5.1. The Mann–Whitney U test used SPSS 22.0, and the data were considered statistically significant if $p < 0.001$.

## Reporting summary

Further information on research design is available in the Nature Portfolio Reporting Summary linked to this article.

## Data availability

All DNA sequencing data in this study was submitted to the Science Data Bank (https://cstr.cn/31253.11.sciencedb.06531; https://doi.org/10.57760/sciencedb.06531), and is publicly available. All other data are available from the corresponding author upon reasonable request.

## Code availability

For data analyses, Python scripts (for layer-by-layer unsupervised pre-training, model structure adjustments, parameter optimization, and model validation) are used (https://github.com/Xyzo21/MNC), as detailed in the "Methods" section of the paper.

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

## Acknowledgements

We are grateful to Pro. Ye Deng (Research Centre for Eco-Environmental Sciences, Chinese Academy of Sciences) for the suggestions and polishing of this article. This work was supported by the Second Tibetan Plateau Scientific Expedition and Research Program (2019QZKK0402, 2019QZKK0307); the National Key Research and Development Program of China (2018YFA0901200); and the Weiqiao-UCAS Special Projects on Low-Carbon Technology Development (GYY-NYHJ-2023-WT-001, GYY-NYHJ-2023-ZY-001).

## Author contributions

Q.Z., A.M., and G.Q.Z.: design. Q.Z., X.C., and X.Z.: methodology. X.C., A.M., and G.L.: sampling. X.N., G.D.Z., and X.C.: investigation and resources. Q.Z. and X.C.: writing of the manuscript with help from A.M., G.Z., and D.F. All authors contributed intellectual input and assistance to this study and manuscript preparation.

## Competing interests
The authors declare no competing interests.
