## [Peer Review File · Communications Biology]

Reviewers' comments:

Reviewer #1 (Remarks to the Author):

The soil carbon sequestration capacity of the Tibetan Plateau grasslands was investigated and it showed an increasing trend under climate warming. By combining measurements of a large number of amino sugars with data on soil microbial communities, carbon-metabolizing function genes, climate parameters and soil properties, the current stock of microbial necromass carbon (MNC) was assessed and the contribution of eco-clusters to MNC was determined. Subsequently, the machine learning model was used to predict the change of MNC storage with climate change. Specifically, climate warming will lead to an average increase of 7.49 mg/g of soil MNC of the Tibetan Plateau grassland. These results are useful for understanding of the overall MNC distribution on the Tibetan Plateau under warming. The results are new and interesting. The manuscript was well prepared. Some expressions in the manuscript need to be modified listed in the following:

Line 24: we divided eco-clusters, calculated their

Line 28: under current and warming conditions

Line 34-37: Two sentences can be merged into one sentence.

Line 65-65: Need rephrase this sentence, based on the above statement.

Line 81: The correlation analysis of beta diversity of dominant and rare microbial communities and microbial necromass carbon is suggested to further illustrate the possible importance of non-dominant microbial community to MNC.

Line 86-87: Abundance of 16S gene copies is not a robust metric for total microbial biomass C.

Line 103-104: Please define "core microbiomes" in methods. Explain in detail the relationship between "core microbiomes" and ecological clusters.

Line 136: Why does the CE model include the first 19 variables? Please explain.

Line 141: K-fold

Line 144-148: I don't consider the improvement between the CE model and the WE model to be vital. It did improve, but the WE model is not far behind.

Line 159-156: I do not understand the connection you're making between microbial consumption of plant C and the consumption of MNC.

Line 173: Why was RCP8.5 chosen? Is the eco-cluster also a key indicator in other RCP scenarios?

Line 235-237: "microbial community changes". Please give an example or explanation of which microorganisms.

Line 252-256: What microorganisms make up the core microbiome and is it similar to the description in line 213-214? Please complete the relevant information.

Line 285-295: Whether this paragraph can be merged with other parts.

Line 376: NEO, <https://neo.sci.gsfc.nasa.gov/>

And some questions:

1. Fig1: The altitude symbology is mostly lost behind the steppe vs. meadow symbology. Altitude is often a poor proxy for climate, so perhaps it's better to just drop it from the figure.
2. What measures were taken to ensure that the previously published data aligned appropriately with the new field data?
3. It can be compared with the driving factors of microbial necromass carbon change in other ecological environments and the climate simulation results, highlighting the important role of the Tibetan Plateau in carbon sequestration under climate warming.

Reviewer #2 (Remarks to the Author):

Zhang et al. have investigated how future climate warming affects microbial necromass carbon sequestration through deep learning model. Their investigation based on large scale sampling and published data. They conducted soil sampling and estimated the MNC by measuring amino sugars, and then they ascertained the soil's capacity for carbon transformation by quantifying the expression of

genes related to carbon degradation and fixation using qPCR. Generally, the study is interesting and the manuscript is well written. I recommend publication of the manuscript after a minor revision to include:

- 1- The term "eco clusters" needs for further explanation in the abstract and introduction sections
- 2- It would be interesting to show data (in the main text) regarding the community composition of NDVI which highly contribute the MNC

Reviewer #3 (Remarks to the Author):

Reviewer #1: The carbon sequestration capacity of the Tibetan Plateau under future warming has been a hot topic of concern in recent years. This article uses modeling to give quantitative conclusions, which is very important. However, I still have some concerns and questions about some of the details of the article

L2: The climate here may be redundant

L28-30: The results and discussion in the article do not quantify the average impact of warming on MNC, how did you arrive at this specific value?

L32: Under climate change conditions is a broad concept and this article should focus on future warming.

L46: Please abbreviate carbon to C, as you have already expressed the abbreviation in the front of the article (L39). All other abbreviated expressions in the article need to be rechecked.

L52: Please abbreviate microbial necromass carbon as MNC, as you have already expressed the abbreviation in the front of the article (L39).

L56-58: Such an assertion seems too absolute, and there have been many large-scale studies in the past.

L199-207: Suggest inserting references here that support the argument.

L237-239: The experimental results do confirm the conclusions of the model fit, but consider whether comparisons with other model simulation results are needed to highlight the sophistication of the model in this article.

Whether the sentences L271-272 and L222-223 are contradictory in the discussion.

L320-326: There seems to be a contradiction here in that the number of sample replicates you collected ($n = 6$) does not match the number of sample replicates in the article you referenced ($n = 7$). (Ding et al)

L326-329: Are the 71 samples here at MNC based on the same experimental methods of detection?

L369-373: There is a large time gap between when data on climate indicators and soil properties are obtained, and I am concerned that using past aridity index to explain current MNC is biased. This may also contribute to the low effect of AI on MNC (Fig.3a).

Fig.1 The samples of microbial necromass carbon (MNC) and microbial related genes don't seem to be perfectly aligned, and I'm confused as to how your linear fit here was achieved?

Fig.3a Here it is proposed to determine the significance of environmental factors on MNC.

Dear Editor and Reviewers:

We are grateful for your insightful and constructive comments on our manuscript entitled "Tibetan Plateau grasslands will increase sequestration of microbial necromass carbon under future warming climate" (COMMSBIO-23-4750-T). We have carefully considered and responded to all reviewers' comments, which help us to improve the manuscript substantially. We have revised the content of the manuscript according to the valuable suggestions from reviewers. All changes in the revised version are highlighted in yellow. The line numbers in our response refer to the revised version of the manuscript. The following are the responses and revisions we have made in response to the reviewers' suggestions on an item-by-item basis. Thanks again to the hard work of the editor and reviewers.

Response to the comments of Reviewer #1:

Comment: The soil carbon sequestration capacity of the Tibetan Plateau grasslands was investigated and it showed an increasing trend under climate warming. By combining measurements of a large number of amino sugars with data on soil microbial communities, carbon-metabolizing function genes, climate parameters and soil properties, the current stock of microbial necromass carbon (MNC) was assessed and the contribution of eco-clusters to MNC was determined. Subsequently, the machine learning model was used to predict the change of MNC storage with climate change. Specifically, climate warming will lead to an average increase of 7.49 mg/g of soil MNC of the Tibetan Plateau grassland. These results are useful for understanding of the overall MNC distribution on the Tibetan Plateau under warming. The results are new and interesting. The manuscript was well prepared. Some expressions in the manuscript need to be modified listed in the following:

Response: We thank the reviewer for the comment and suggestion. We have carefully considered all the comments and responded one by one. As follows:

Comment: 1. Line 24: we divided eco-clusters, calculated their

Response: Done accordingly on lines 24-25.

Comment: 2. Line 28: under current and warming conditions

Response: Done accordingly on line 29.

Comment: 3. Line 34-37: Two sentences can be merged into one sentence.

Response: Thanks for the suggestion. This sentence was revised in the new manuscript of line 35-38.

Comment: 4. Line 65-65: Need rephrase this sentence, based on the above statement.

Response: Thanks for the suggestion. Based on the suggestion, this sentence was revised in the new manuscript of line 66-69.

Comment: 5. Line 81: The correlation analysis of beta diversity of dominant and rare microbial communities and microbial necromass carbon is suggested to further illustrate the possible importance of non-dominant microbial community to MNC.

Response: Thanks for the comment. Based on the comment of the reviewer, we conducted a correlation analysis between the beta diversity of dominant and rare microbial communities and microbial necromass carbon (MNC) (Fig. 1). The results showed that there was a certain relationship between the community structure and MNC, in which the dominant microbial communities were significantly positively correlated with MNC, while the rare microbial communities were negatively correlated with MNC. However, the fitting effect between dominant microbial communities and MNC ($R^2 = 0.42, p = 0.0004$) was significantly better than that between rare microbial communities and MNC ($R^2 = 0.17, p = 0.0390$), indicating the contribution of dominant microbial communities to MNC and laying a foundation for subsequent research. We have described the relevant results in the new manuscript of lines 96-99, and the pictures (Fig. S3) can be found in the supplementary materials.

Fig. 1 The correlation analysis of PC1 of dominant (A) and rare (B) microbial communities and microbial necromass carbon (MNC).

Comment: 6. Line 86-87: Abundance of 16S gene copies is not a robust metric for total microbial biomass C.

Response: We thank the reviewer for the comment and suggestion. According to published studies, 16S rRNA gene copies can evaluate microbial abundance, indicating a certain linear relationship between the abundance of 16S gene copies and total microbial biomass C. Therefore, the abundance of 16S gene copies can be used as a characterization indicator of total microbial biomass C because it is present in all prokaryotic bacteria. Some published papers have explored and supported the relationship between 16S rRNA gene copies/DNA concentration and total microbial biomass to some extent, and these findings jointly strengthen the application of this indicator in related studies. In sum, abundance of 16S gene copies can reflect or characterize the total microbial biomass C.

Reference:

1. Fierer, N., et al. (2007). Toward an ecological classification of soil bacteria. *Ecology*, 88(6), 1354-1364.
2. Kembel, S.W., et al. (2012). Incorporating 16S gene copy number information improves estimates of microbial diversity and abundance. *PLOS Computational Biology*, 8(10), e1002743.

3. Liang, C., et al. (2017). The importance of anabolism in microbial control over soil carbon storage. *Nature Microbiology*, 2, 17105.

4. Chen, Y.L., et al. (2020). Large-scale evidence for microbial response and associated carbon release after permafrost thaw. *Global Change Biology*, 27, 3218-3229.

5. Gong, H.Y., et al. (2021). Soil microbial DNA concentration is a powerful indicator for estimating soil microbial biomass C and N across arid and semi-arid regions in northern China. *Applied Soil Ecology*, 160, 103869.

Comment: 7. Line 103-104: Please define “core microbiomes” in methods. Explain in detail the relationship between “core microbiomes” and ecological clusters.

Response: Thanks for this helpful comment. In order to facilitate reader understanding and make the article clearer, we have replaced “core microbiomes” with “dominant phylotypes” in the new manuscript of line 25, line 108, line 260, line 261, line 264, line 267, line 313, and line 421. The definition of dominant phylotypes has been explained and defined in the method of line 418-421. In addition, ecological clusters are composed of dominant phylotypes in the new manuscript of line 415-430.

Comment: 8. Line 136: Why does the CE model include the first 19 variables? Please explain.

Response: Thanks for the comment. In this study, a total of 28 variables were evaluated using random forest (RF), and the first 19 variables accounted for 70% of the total variables, which have been selected and explained in the manuscript of lines 142-146. Among them, 70% is not a definite or constant parameter, but the first 19 variables are comprehensively selected according to experience and reference to the impact of environmental factor variables on MNC in other studies. For example, NPP and TN, which rank relatively low among variables, have been reported in other studies to have an important impact on MNC. Therefore, in this study, we also included them as input variables for the model. The principle of variable screening is to retain as many variables as possible that have a significant impact on the MNC, so as to ensure that some key influencing factors are not missed. Don't worry too much about redundancy

in variable selection, because the deep learning model itself can learn to optimize the input variables to ensure the accuracy of the output results.

Comment: 9. Line 141: K-fold

Response: Done accordingly on line 146.

Comment: 10. Line 144-148: I don't consider the improvement between the CE model and the WE model to be vital. It did improve, but the WE model is not far behind.

Response: We thank the reviewer for the comment and suggestion. The training model should first ensure the rationality of the model structure. In this study, the appropriate model structure was determined after several pre-trainings, and then the CE (including all 19 variables) and WE (not contain the HMAP&LMAT, Hele, HSOC<N, Lthickness, and Hele&LNDVI.) models were defined by changing the input variables (lines 142-146). Essentially, the structure of the CE model is the same as that of the WE model, and the difference is the input variables. Therefore, it is not surprising that there is little difference between the prediction results of the WE and CE model because the rationality of the model structure leads to the excellent simulation effect of the two models. However, the CE model ($R^2 = 0.82$, RMSE = 4.16) has better simulation performance than the WE model ($R^2 = 0.75$, RMSE = 4.91) of lines 147-149. In addition, the CE model's better simulation results result from the average of multiple simulation results. The hold-out validation result of the WE model showed in Fig. 2 below (in the new manuscript of Fig. 3). The result shows that the hold-out validation of the CE model is better than that of the WE model in the new manuscript of lines 151-152.

Fig. 2 **RF analysis results and model validation.** (a) Importance ranking of RF analysis between MNC and environmental variables. (b) CE model K-fold cross validation result. (c) WE model K-fold cross validation result. (d) CE model hold-out validation result. (e) WE model hold-out validation result.

Comment: 11. Line 159-156: I do not understand the connection you're making between microbial consumption of plant C and the consumption of MNC.

Response: Thanks for the comment. Plant C and subsequent generation have a certain influence on the consumption of MNC by microorganisms. Generally, plant C promoted the formation of stubborn carbon or stable carbon after the action of microorganisms. For example, Liang et al. explored how microbial processes convert plant C into more recalcitrant forms, including microbial necromass, which can be an important carbon sink. Kallenbach et al. showed how microbial consumption of plant-derived carbon can contribute to the formation of stable organic matter, including microbial necromass. Cotrufo et al. explored the various pathways of plant litter decomposition, including the transformation into microbial biomass and ultimately into more stable forms of soil carbon, like microbial necromass. Thus, it is necessary to discuss the connection between microbial consumption of plant C and the consumption of MNC (lines 202-210 and lines 248-253).

Reference

Liang, C., et al. (2011). Microbial production of recalcitrant organic matter in

global soils: implications for productivity and climate policy. *Nature Reviews Microbiology*, 9(1), 75-79.

Kallenbach, C.M., et al. (2016). Direct evidence for microbial-derived soil organic matter formation and its ecophysiological controls. *Nature Communications*, 7(1), 1-10.

Cotrufo, M.F., et al. (2015). Formation of soil organic matter via biochemical and physical pathways of litter mass loss. *Nature Geoscience*, 8(10), 776-779.

Comment: 12. Line 173: Why was RCP8.5 chosen? Is the eco-cluster also a key indicator in other RCP scenarios?

Response: Thanks for the comment. RCP8.5 is the most severe scenario for climate warming. It is chosen as the background for warming because it can maximize the impact of climate warming on MNC and may make the changes to MNC more obvious. In addition, we also predicted the MNC of alpine meadow and alpine steppe under the RCP4.5 scenario. The results showed that the average predicted MNC under the RCP4.5 scenario was 16.92 mg/g, while the average predicted MNC under the RCP8.5 scenario was 18.45 mg/g. Moreover, the predicted MNC values of alpine meadows and alpine steppe under the RCP8.5 scenario were higher than those under the RCP4.5 scenario (Fig. 3). Based on the research objectives of this article, we have chosen RCP8.5 as a more appropriate option to highlight the importance of warming on MNC and the research topic.

Fig. 3 Predicted MNC of alpine meadow and alpine steppe under RCP4.5 and RCP8.5 scenarios.

The indicator effect of the relative abundance of eco-cluster on MNC is based on the investigation results of existing data and is consistent with the objective facts of MNC distribution in alpine grassland and meadow at present. Therefore, the eco-cluster, as a predictor, is the premise of the model prediction hypothesis, and of course, it can also be used to simulate the distribution changes of MNC under other RCP scenario models.

Comment: 13. Line 235-237: “microbial community changes”. Please give an example or explanation of which microorganisms.

Response: Thank you for the insightful comment. Corresponding to the previous sentence in the manuscript, Table S3 shows the changes of eco-cluster, which corresponds to the changes in the overall distribution pattern of the microbial community, not that one microbial community changes. According to the reviewer's suggestion, we made a modification in the new manuscript of line 243.

Comment: 14. Line 252-256: What microorganisms make up the core microbiome and is it similar to the description in line 213-214? Please complete the relevant information.

Response: Thanks for the comment. Core microbiome is composed of OTUs with an abundance of over 10% in the sample. In order to make the article description clearer and help readers understand this study, we have uniformly replaced core microbiome with dominant phylotypes in the new manuscript and described them. Line 252-256 (in the new manuscript of lines 260-264) and line 213-214 (in the new manuscript of line 219) are consistent. More explanation and elaboration on the relationship between dominant phylotypes and eco-cluster have been provided in the new manuscript of lines 415-430.

Comment: 15. Line 285-295: Whether this paragraph can be merged with other parts.

Response: We thank the reviewer for the comment and suggestion. Based on the reviewer's suggestion, we have made revisions to make the logic more concise and clear in the new manuscript of lines 295-297 and lines 316-326.

Comment: 16. Line 376: NEO, <https://neo.sci.gsfc.nasa.gov/>

Response: Done accordingly on line 386.

Comment: 17. Fig1: The altitude symbology is mostly lost behind the steppe vs. meadow symbology. Altitude is often a poor proxy for climate, so perhaps it's better to just drop it from the figure.

Response: Thanks for the suggestion. Based on the reviewer's suggestions, we have made modifications to Fig. 1 in the new manuscript of line 100. The new Fig. 1 in the manuscript is shown in the following Fig. 4.

Fig. 4 The new Fig. 1

Comment: 18. What measures were taken to ensure that the previously published data aligned appropriately with the new field data?

Response: Thank you for the insightful comment. In order to make our study more scientific and reliable, we have carried out a detailed description in the manuscript method, including the consistency of sampling method, sample testing method and sequencing data analysis method in the new manuscript of lines 329-334, lines 344-353, and lines 398-414.

Comment: 19. It can be compared with the driving factors of microbial necromass carbon change in other ecological environments and the climate simulation results, highlighting the important role of the Tibetan Plateau in carbon sequestration under climate warming.

Response: Thanks for the comment. In this study, we mainly focus on the alpine grassland and meadow of the Tibetan Plateau, with less attention paid to other ecosystems. However, existing studies have not simulated and predicted MNC under climate warming conditions in either alpine ecosystems or other ecosystems. The next step can be to attempt to simulate and predict MNC under warming conditions in other ecosystems. In addition, the important carbon sequestration effect of the Tibetan Plateau has been described in detail in the introduction (lines 35-41).

Response to the comments of Reviewer #2:

Comment: Zhang et al. have investigated how future climate warming affects microbial necromass carbon sequestration through deep learning model. Their investigation based on large scale sampling and published data. They conducted soil sampling and estimated the MNC by measuring amino sugars, and then they ascertained the soil's capacity for carbon transformation by quantifying the expression of genes related to carbon degradation and fixation using qPCR. Generally, the study is interesting and the manuscript is well written. I recommend publication of the manuscript after a minor revision to include:

Response: We thank the reviewer for the comment and suggestion. We have carefully considered all the comments and responded one by one. As follows:

Comment: 1. The term “eco clusters” needs for further explanation in the abstract and introduction sections.

Response: Thanks for the comment. According to the reviewer's suggestion, we have described and explained in the abstract and introduction of the new manuscript of lines 24-25 and lines 75-77.

Comment: 2. It would be interesting to show data (in the main text) regarding the community composition of NDVI which highly contribute the MNC.

Response: We thank the reviewer for the comment and suggestion. The reviewer's suggestions further support our interpretation of the study (Fig. 5). We have described and discussed the relevant results in the new manuscript of lines 283-285, and the pictures (Fig. S7) can be found in the supplementary materials.

Fig. 5 The correlation analysis of NDVI and MNC.

Response to the comments of Reviewer #3:

Comment: The carbon sequestration capacity of the Tibetan Plateau under future warming has been a hot topic of concern in recent years. This article uses modeling to give quantitative conclusions, which is very important. However, I still have some concerns and questions about some of the details of the article

Response: We thank the reviewer for the comment and suggestion. We have carefully considered all the comments and responded one by one. As follows:

Comment: 1. L2: The climate here may be redundant

Response: Done accordingly on line 2.

Comment: 2. L28-30: The results and discussion in the article do not quantify the average impact of warming on MNC, how did you arrive at this specific value?

Response: Thank you very much for this comment. This is a calculation result. Specifically, lines 189-190 “The mean MNC value increases from 10.96 mg/g to 18.45 mg/g under simulated warming conditions (Fig. 4d).” corresponds to line 30 “with an average increase of 7.49 mg/g, a 68.3% increase”.

Comment: 3. L32: Under climate change conditions is a broad concept and this article should focus on future warming.

Response: Thanks for the comment. Based on the suggestion of the reviewer, we have made revisions in the new manuscript of line 33.

Comment: 4. L46: Please abbreviate carbon to C, as you have already expressed the abbreviation in the front of the article (L39). All other abbreviated expressions in the article need to be rechecked.

Response: Thank you very much for this comment. Done accordingly on line 40, line 46, line 47, line 78, line 79, line 233, line 249, line 268, line 286, line 288, line 289 line 290, line 293, and line 300.

Comment: 5. L52: Please abbreviate microbial necromass carbon as MNC, as you have already expressed the abbreviation in the front of the article (L39).

Response: Thank you very much for this comment. Done accordingly on line 52.

Comment: 6. L56-58: Such an assertion seems too absolute, and there have been many large-scale studies in the past.

Response: Thank you very much for this comment. According to the reviewer's comments, we have re-described this sentence in the new manuscript of lines 56-59.

Comment: 7. L199-207: Suggest inserting references here that support the argument.

Response: Thank you for your comment. According to the comments of the reviewer, we introduced some papers to support our study and discuss the results. (Cotrufo MF, Soong JL, Horton AJ, Campbell EE, Haddix ML, Wall DH, Parton WJ. Formation of soil organic matter via biochemical and physical pathways of litter mass loss. *Nat Geosci.* 2015; 8(10):776-779; Louca S, Polz MF, Mazel F, et al. Function and functional redundancy in microbial systems. *Nat Ecol Evol.* 2018; 2(6):936-943. In the new manuscript of reference #16 and #17.)

Comment: 8. L237-239: The experimental results do confirm the conclusions of the model fit, but consider whether comparisons with other model simulation results are needed to highlight the sophistication of the model in this article.

Response: Thanks for the comment. The reviewer's suggestion is very meaningful, which contributes to the stability and scientificity of the model and prediction results and also provides an effective evaluation of the model used in this paper. However, there are currently no model simulation and prediction results for MNC in the Qinghai Tibet Plateau, so an effective comparison cannot be made.

Comment: 9. Whether the sentences L271-272 and L222-223 are contradictory in the discussion.

Response: Thanks for the comment. These two sentences are not contradictory. L222-223 (in the new manuscript of lines 229-230) describes that the impact of bacterial communities on MNC is higher (RF ranking higher) because bacterial communities respond more strongly to climate warming than fungi and can better indicate changes in MNC. It is emphasized here that bacterial communities play an important role in predicting MNC. L271-272 (in the new manuscript of lines 280-281) describes that the proportion of fungal residues in MNC is higher than that of bacterial residues, and they are more likely to accumulate in acidic soils such as AM. The reason why MNC can accumulate more in AM than in AS is emphasized and explained here. Therefore, these two sentences describe different issues, so they are not contradictory.

Comment: 10. L320-326: There seems to be a contradiction here in that the number of sample replicates you collected ($n = 6$) does not match the number of sample replicates in the article you referenced ($n = 7$). (Ding et al)

Response: Thank you very much for this comment. The sampling method (lines 331-333) used in this study is the same as in previous studies (Ding, J. et al. The permafrost carbon inventory on the Tibetan Plateau: a new evaluation using deep sediment cores. *Global Change Biology* 22, 2688-2701 (2016).), except that the number of parallel samples is different. The data used is obtained by taking the average value in this work.

Comment: 11. L326-329: Are the 71 samples here at MNC based on the same experimental methods of detection?

Response: Thank you very much for your suggestion. In order to make the data comparable, the same amino sugar determination method was used. The method has been described in the manuscript of lines 398-414, and the keyword used for searching data is based on the "amino sugar" assay (line 338).

Comment: 12. L369-373: There is a large time gap between when data on climate indicators and soil properties are obtained, and I am concerned that using past aridity index to explain current MNC is biased. This may also contribute to the low effect of AI on MNC (Fig.3a).

Response: Thanks for the comment. This database is commonly used in ecological research (72,202 downloads and 247 citations, https://figshare.com/articles/dataset/Global_Aridity_Index_and_Potential_Evapotranspiration_ET0_Climate_Database_v2/7504448/3), and the vast majority of studies use this database for AI assessment. In this study, we used the average of AI over a larger time scale to minimize errors. The reviewer's suggestion is very valuable, so we also combined the other research results of the papers when we comprehensively considered the environmental factors that may affect MNC, and discussed the situation of AI contributing MNC in the new manuscript of lines 270-282. See reference (Trabucco, Antonio; Zomer, Robert (2019). Global Aridity Index and Potential Evapotranspiration (ET0) Climate Database v2. figshare. Dataset. <https://doi.org/10.6084/m9.figshare.7504448.v3>) for more information on the use of AI in the lines 381-383.

Comment: 13. Fig.1 The samples of microbial necromass carbon (MNC) and microbial related genes don't seem to be perfectly aligned, and I'm confused as to how your linear fit here was achieved?

Response: Thanks for the comment. In Fig.1, the fitting curve here is only for displaying a trend, not for a very perfect fit. In addition, the vast majority of samples are within a reliable threshold range of 95%, and the fitting results are significant. Therefore, our results are reliable.

Comment: 14. Fig.3a Here it is proposed to determine the significance of environmental factors on MNC.

Response: Thank you for your suggestions and comments. Yes, Fig.3a shows the use of random forests (RF) to identify important environmental factors for MNC, as described earlier (Delgado-Baquerizo, M. et al. A global atlas of the dominant bacteria found in soil. *Science* 359, 320-325 (2018).).

We thank you for your time and consideration of our manuscript.

Yours sincerely,

Anzhou Ma

Email: azma@rcees.ac.cn

Phone: 86-010-6284-9156

Research Center for Eco-Environmental Sciences, Chinese Academy of Sciences,
Beijing, 100085, China

REVIEWERS' COMMENTS:

Reviewer #1 (Remarks to the Author):

The authors have revised the manuscript accordingly. And the revisions are acceptable. It can be accepted in current state.

Reviewer #3 (Remarks to the Author):

The changes imposed in the first revision make the paper acceptable for publication.